# Impact of Evolution of Self-Expandable Aortic Valve Design: Peri-Operative and Short-Term Outcomes

**DOI:** 10.3390/jcm12051739

**Published:** 2023-02-21

**Authors:** Evangelia Bei, Vasileios Voudris, Konstantinos Kalogeras, Evangelos Oikonomou, Ioannis Iakovou, Ilias Kosmas, Charalampos Kalantzis, Michael-Andrew Vavuranakis, Panteleimon Pantelidis, George Lazaros, Dimitrios Tousoulis, Constantinos Tsioufis, Manolis Vavuranakis

**Affiliations:** 1First Department of Cardiology, Hippokration Hospital, Medical School, National and Kapodistrian University of Athens, 11528 Athens, Greece; 2Interventional Department of Cardiology, Onassis Cardiac Surgery Center, 17674 Athens, Greece; 3Third Department of Cardiology, Sotiria General Hospital for Chest Diseases, Medical School, National and Kapodistrian University of Athens, 11527 Athens, Greece

**Keywords:** TAVI, implantation depth, paravalvular leak, permanent pacemaker implantation, aortic valve stenosis

## Abstract

A few data exist on the differences of implantable aortic valve bio-prostheses. We investigate three generations of self-expandable aortic valves in terms of the outcomes. Patients undergoing transcatheter aortic valve implantation (TAVI) were allocated into three groups according to the valve type: group A (CoreValve^TM^), group B (Evolut^TM^R) and group C (Evolut^TM^PRO). The implantation depth, device success, electrocardiographic parameters, need for permanent pacemaker (PPM), and paravalvular leak (PVL) were assessed. In the study, 129 patients were included. The final implantation depth did not differ among the groups (*p* = 0.07). CoreValve^TM^ presented greater upward jump of the valve at release (2.88 ± 2.33 mm vs. 1.48 ± 1.09 mm and 1.71 ± 1.35 mm, for groups A, B, and C, respectively, *p* = 0.011). The device success (at least 98% for all groups, *p* = 1.00) and PVL rates (67% vs. 58%, vs. 60% for groups A, B, and C, respectively, *p* = 0.64) did not differ. PPM implantation within 24 h (33% vs. 19% vs. 7% for groups A, B, and C, respectively, *p* = 0.006) and until discharge (group A: 38% vs. group B: 19% and group C: 9%, *p* = 0.005) was lower in the newer generation valves. Newer generation valves present better device positioning, more predictable deployment, and fewer rates of PPM implantation. No significant difference in PVL was observed.

## 1. Introduction

Transcatheter aortic valve implantation (TAVI) is the treatment option for patients with severe aortic stenosis who are considered to be inoperable or present with a high surgical risk [1,2]. With the population that is eligible for transcatheter treatment expanding over the past years [3,4,5], extensive research and technological advancements have improved the design, functionality, and clinical outcomes of self-expandable valves. Improvements are focused on minimizing the risk of mal-positioning, paravalvular leak (PVL) and vascular complications, as well as lowering the need for post-procedure new pacemaker implantation [6]. While the first-generation self-expandable CoreValve^TM^ was non-retrievable, the second-generation Evolut^TM^R was repositionable and recapturable, with an expanding height of the pericardial skirt. Finally, the third-generation Evolut^TM^PRO, on top of the previous features, has an outer porcine pericardial wrap on the lower part of stent frame in order to increase the surface contact and reduce the PVL [7].

Although several studies address the differences of certain transcatheter aortic valve bio-prostheses, a few data exist to compare the procedural performance and clinical outcomes of the three sequential versions of the Medtronic CoreValve device [8]. Therefore, we compare the immediate and short-term clinical outcomes of three generations of self-expandable aortic valves in patients that underwent TAVI.

## 2. Materials and Methods

### 2.1. Study Design and Participants

Consecutive patients with severe aortic stenosis who underwent TAVI between January 2012 and July 2019 in two experienced centers were retrospectively analyzed. Only the patients treated with the Medtronic system valves (Medtronic, Minneapolis, MN, USA) via transfemoral access were included in the analysis. The patients who presented with unicuspid or bicuspid aortic valves and/or a history of previous aortic valve replacement (AVR) were excluded. Patients with a pre-procedural heart rhythm other than the sinus rhythm were also excluded.

The patients were allocated into three groups according to the type of implanted self-expandable bioprosthesis: group A—CoreValve™; group B—Evolut™R; group C—Evolut^TM^PRO. CoreValve^TM^ was used after receiving CE (Conformité Européenne) Mark in 2007, Evolut^TM^R was implanted in October 2014, and Evolut^TM^PRO was implanted from October 2017 to May 2019. Using the Evolut^TM^PRO group as a reference for the propensity score matching method, forty-three patients were finally selected and included in each group.

### 2.2. Procedural Assessment and Patient Follow-Up

All of the patients underwent a pre-procedural screening evaluation including electrocardiogram (ECG), transthoracic echocardiography (TTE), coronary angiography accompanied by reperfusion if needed, and multi-slice computed tomography (MSCT). The anatomical parameters in MSCT were evaluated by 3mensio Valves (version 9.1.sp3, 3mensio Pie Medical Imaging B.V., Maastricht, The Netherlands). The calcium score was measured manually using the same program by using a cut-off point of 450 Housefield Units (HU) in the non-contrast scans in order to detect all of the calcified areas on the aortic annulus area [9,10]. During the procedure, pre-dilatation was performed at the discretion of the attending physician. At the end of the procedure, ascending aortic angiography was carried out after valve release and the withdrawal of the delivery system to evaluate the final depth and estimate eventual para valvular regurgitation. Post-dilatation was conducted selectively in patients with device deformation and moderate or severe PVL values. Valve implantation was performed by the same operator in each center.

Implantation depth (ID) was defined as the distance (in millimeters, mm) from the non-coronary cusp (NCC) to the deepest end of the expanded bioprosthesis into the left ventricle after its final release. Bioprosthesis “jump” was defined as the difference in distance prior to the disengagement of the bioprosthesis and the final position after its disengagement from the delivery system (difference between the initial position of bioprosthesis and the final ID of the valve after deployment). Device success was defined according to Valve Academic Research Consortium (VARC-3) criteria [11], whereas PVL was assessed with angiography according to the Sellers criteria [12]. From grades 2 to 4, valve regurgitation was considered to be significant, while no regurgitation and grade 1 indicated an optimal result. An electrocardiogram was performed daily after the procedure. The need for permanent pacemaker immediately post-TAVI and until discharge was defined by a complete or high-degree atrio-ventricular blockage persisting for 24–48 h after implantation, a new onset alternating bundle branch blockage, a pre-existing right bundle branch blockage with new conduction disturbance, a left bundle branch blockage with a QRS duration > 150 ms or PR interval prolongation >240 ms with ongoing prolongation [13]. TTE was performed immediately after TAVI, and also after 24 and 48 h, as well as at the time of the patient’s discharge. Clinical and echocardiographic data were recorded at 1 and 6 months post-procedurally on an outpatient basis.

### 2.3. Outcomes and Extracted Data

Demographic data and clinical, echocardiographic, and electrocardiographic parameters at the baseline and during the follow-up were collected. The primary outcomes were the implantation depth, jump of bioprosthesis, and device success according to the VARC-3 criteria [13]. Secondary outcomes were conduction system abnormalities and the need for permanent pacemaker implantation after TAVI, as well as PVL after implantation, at discharge, and at 1 and 6 month of follow-up.

### 2.4. Statistical Analysis

Propensity scores were used, considering age, sex, MSCT-derived anatomical characteristics, and the CT-guided valve size selection of group C, as matching variables. Distributions of the baseline characteristics and outcomes are described as their mean and standard deviation (SD) (for continuous variables) or as median with the interquartile range when they are not normally distributed. Categorical variables are presented as frequencies and percentages. Normality was assessed by Shapiro–Wilk test. For the variables that were normally distributed, a parametric test in order to explore any differences was applied. For the not normally distributed variables, a non-parametric test was performed. The three groups were compared with an analysis of variance (ANOVA) with a correction using Tukey’s HSD or with Kruskal–Wallis test for the not normally distributed variables. Moreover, differences in the outcome rates were tested with Pearson’s chi-squared or Fisher’s exact tests. The significance threshold was set to 0.05. All of the analyses were performed in SPSS 25 (Armonk, NY, USA: IBM Corp).

## 3. Results

### 3.1. Patient Characteristics

In total, 400 patients with severe aortic stenosis underwent TAVI in both centers. Overall, one hundred and sixty (160) patients met the eligibility criteria. Of them, 129 were evenly distributed to each group (43 subjects each) after propensity matching (Figure 1).

The mean age was 81.14 ± 5.99 years, while 59 (46%) patients were males. The baseline demographic, anatomical, echocardiographic, and clinical characteristics were similar among the groups (Table 1). The pre-procedural electrocardiographic findings did not significantly differ, except for the pre-existing first degree atrioventricular (AV) blockage, which was more commonly observed in group B (26% vs. 5% for group A vs. 14% for group C, *p* = 0.03, with A vs. B, *p* = 0.01, A vs. C, *p* = 0.17 and B vs. C, *p* = 0.18).

### 3.2. Peri-Procedural Outocomes

As seen in Table 2, the implantation depths (ID) were 3.63 ± 2.77 mm for group A, 4.96 ± 1.90 mm for group B, and 4.44 ± 2.24 mm for group C (*p* = 0.07). Interestingly, group A (CoreValve^TM^ valve) presented with a significantly higher ID than group B did (Evolut^TM^R, *p* = 0.02). However, the bioprosthesis jump was greater in group A compared to those in groups B and C (2.88 ± 2.33 mm vs. 1.48 ± 1.09 mm vs. 1.71 ± 1.35 mm, *p* = 0.01, with A vs. B, *p* < 0.01, A vs. C, *p* = 0.03 and B vs. C, *p* = 0.57). Accordingly, the patients with first-generation valves (group A) demonstrated higher rates of ID, which were above the level of annulus than those of group B (23% vs. 5% vs. 9%, for groups A, B and C, respectively, *p* = 0.02, with A vs. B, *p* = 0.01, A vs. C, *p* = 0.08 and B vs. C, *p* = 0.4). Nevertheless, a considerable disparity among the groups in terms of normal ID was not observed (54% vs. 65% vs. 58%, for groups A, B, and C, respectively, *p* = 0.54). Balloon pre-dilatation was remarkably less common among the patients treated with the newer generation valves (group A: 79.1% vs. group B: 20.9% vs. group C: 16.3%, *p* < 0.001, with A vs. B, *p* < 0.001, A vs. C, *p* < 0.001 and B vs. C, *p* = 0.58).

Upon completion of the TAVI procedure, but before post-balloon dilatation, the PVL rates of grade 1 were similar among the groups (group A: 67% vs. group B: 58% vs. group C: 60%, *p* = 0.64). There was no difference in the device success rate between the groups. Post-dilatation was more frequent in the second-generation valves (group A: 23% vs. group B: 42% vs. group C: 30%, *p* = 0.17, with A vs. B, *p* = 0.07, A vs. C, *p* = 0.47 and B vs. C, *p* = 0.26).

### 3.3. Echocardiographic Characteristics at Discharge and Follow-Up

The post-procedural, pre-discharge, and follow-up echocardiography did not show any difference among the groups regarding the PVL grade ≤1 at discharge (group A: 87% vs. group B: 92% vs. group C: 88%, *p* = 0.79), after one month (group A: 94% vs. group B: 86% vs. group C: 88%, *p* = 0.62), and after six months (group A: 90% vs. group B: 94% vs. group C: 78%, *p* = 0.19).

Furthermore, a large proportion of patients with grade 2 PVL at the end of TAVI presented with an improvement to grade ≤1 at the point of discharge (group A: 35% vs. group B: 36% and group C: 40%, *p* = 0.049), favoring the newer generation valves. Moreover, in comparing the PVL grades from one to six months, no statistical difference in the proportions of PVL reduction was observed (group A: 4% vs. group B: 13% vs. group C: 7%, *p* = 0.54). At the point of discharge, no differences were shown for the peak aortic valve gradient (group A: 17.80 ± 9.11 mmHg vs. group B: 15.71 ± 8.01 mmHg vs. group C: 16.43 ± 9.13 mmHg, *p* = 0.55), mean aortic valve gradient (group A: 9.46 ± 4.98 mmHg vs. group B: 8.11 ± 4.73 mmHg vs. group C: 10.22 ± 5.99 mmHg, *p* = 0.31), and the peak aortic valve velocity (group A: 2.07 ± 0.45 m/sec vs. group B: 1.90 ± 0.50 m/sec vs. group C: 1.99 ± 0.51 m/sec, *p* = 0.34) among the three groups.

### 3.4. Electrocardiographic Characteristics

Newer generation valves exhibited lower rates of electrocardiographic changes (Table 3). A lower rate of permanent pacemaker implantation at 24 h, due to third degree blockage or advanced AV blockage, was observed for the newer valves (group A: 33% vs. group B: 19% vs. group C: 7%, *p* = 0.01, with A vs. B, *p* = 0.14, A vs. C, *p* < 0.01 and B vs. C, *p* = 0.11), which remained until the point of discharge (group A: 38% vs. group B: 19% vs. group C: 9%, *p* < 0.01, with A vs. B, *p* = 0.05, A vs. C, *p* < 0.01 and B vs. C, *p* = 0.21). The CoreValve^TM^ group developed AV blockages more often immediately after the procedure (group A: 33% vs. group B: 12% vs. group C: 12%, *p* = 0.02, with A vs. B, *p* = 0.02, A vs. C, *p* = 0.02 and B vs. C, *p* = 1.00).

## 4. Discussion

This study emphasizes the differences between different generations of self-expandable valves. While, better and more stable positioning was shown for newer devices, along with lower rates of adverse electrocardiographic changes or the reduced need for permanent pacemaker implantation, a statistical significance was not reached for PVL reduction.

In detail, the next-generation valves demonstrated better performances in terms of stability, with lower rates of translocation and over the annulus final implantation as compared to those of the first-generation ones. Although many studies have implicated several factors such as anatomical or morphological and peri-procedural characteristics that may affect implantation depth [14], the design of bioprosthesis appears to be a considerable parameter. Indeed, newer generation valves have the advantage of retrieving and recapturing the bioprosthesis, with the control of the ideal final implantation depth remaining under the control of the operator. Furthermore, improved maneuverability and the less bulky delivery system in the newer generation valves contributes to the more predictable position of the bioprosthesis. Despite this feature and the manufacturing modifications in second- and third-generation bio-prostheses, the device radial force remains a factor that affects the positioning.

Concerning factors that usually affect paravalvular regurgitation, anatomical, and morphological factors such as calcification, inappropriate valve size, mal-positioning, and the implantation depth of the bioprosthesis have been shown to play a significant role [12]. Having minimized anatomical and morphological differences between the studied groups, this study revealed that the first-generation CoreValve^TM^ presented less PVL improvement between post-procedural evaluation and the point of discharge, as well as between the point of discharge and 1 month of follow-up, compared to those of the newer generation valves. This is explicable by its design with a pericardial wrap in Evolut^TM^PRO, targeting the minimization of PVL after TAVI [7,15]. It is possible that more time is needed for the new pericardial wrap to expand effectively in addition to the increased radial force that characterizes newer generation devices. Nevertheless, in the current study, we found that the Evolut^TM^PRO group seems to have comparable results in PVL with those of the oldest generation valves at the point of discharge, 1, and 6 months of follow-up. Moreover, our findings are unlikely to have been affected by the calcium score since we did not find differences among the three study groups in terms of the calcium score.

To reduce the PVL after TAVI, improvements in the design of the bioprosthesis and also in the implantation technique have taken place, which may affect the conduction system. Additionally, contact pressure by the device frame to the adjacent anatomical structures at the level of annulus and LVOT [16], along with the length of the membranous septum, can increase the risk of conduction abnormalities, which may require permanent pacemaker implantation [17]. In our study, the findings are comparable to those of the Medtronic TAVR 2.0 US Clinical Study [15] regarding permanent pacemaker implantation events within the first day after TAVI. In detail, the CoreValve^TM^ seems to induce a greater need for pacemaker implantation compared to that of Evolut^TM^R and PRO between the first day and the end of the study. Aligning with our results, the ATLAS registry showed that the attachment of the outer pericardial wrap of the Evolut^TM^PRO did not seem to increase the need for new pacemaker implantation compared with that of Evolut^TM^R [18]. This may be explained by the fact that extra pericardial wrap is not designed to exert extra force to LVOT and may reduce the direct mechanical interaction of the device [15]. Moreover, the experience of the operator and the more stable position/less translocation of newer generation valves may account for lower rates of conduction system damage [19,20]. Furthermore, adjustments in the context of the evolution of the TAVI technique, such as the cusp overlap technique could also result in the reduction of conduction disturbances [21]. Nevertheless, the higher target implantation depth in newer generation devices may pose further difficulties regarding future coronary interventions [22,23].

## 5. Conclusions

The design evolution of self-expandable bioprosthetic aortic valves seems to positively affect the peri-procedural outcomes, such as the easier and more predictable valve positioning of the next-generation valves. From a clinical viewpoint, this is translated into the optimal implantation of the bioprosthesis, in addition to less conduction abnormalities and lower rates of permanent pacemaker implantation after TAVI. On the other hand, despite the modifications and the addition of an outer pericardial tissue wrap in the next-generation bioprosthetic valves, PVL was not significantly reduced.

## Figures and Tables

**Figure 1 jcm-12-01739-f001:**
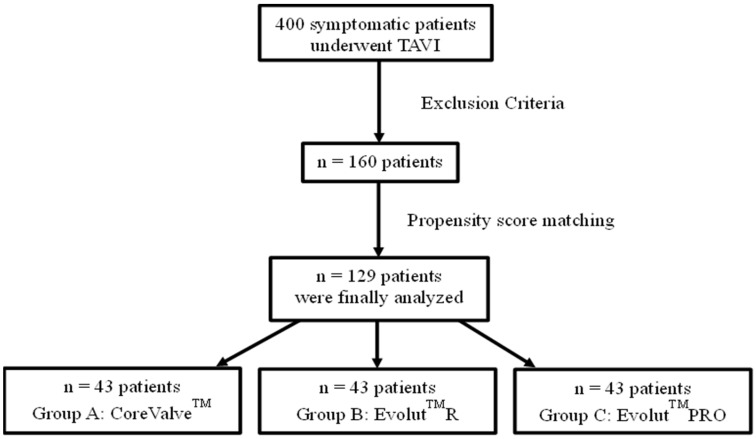
Flow diagram of the study groups.

**Table 1 jcm-12-01739-t001:** Baseline characteristics of study groups.

Characteristic	CoreValve^TM^	Evolut^TM^R	Evolut^TM^PRO	*p* Value
Patients, No	43	43	43	-
Age, years	79.79 ± 5.54	81.19 ± 6.54	82.47 ± 5.70	0.06
Gender (male, %)	51	40	49	0.52
Body Mass Index, (kg/m^2^)	27.40 ± 5.61	27.48 ± 3.94	28.19 ± 4.87	0.17
Hypertension, (%)	86	86	93	0.48
Diabetes mellites, (%)	33	30	33	0.94
Dyslipidemia, (%)	65	72	67	0.78
Coronary artery disease, (%)	36	56	44	0.17
Previous cardiac surgery, (%)	21	26	22	0.72
Peripheral artery disease, (%)	42	37	23	0.18
NYHA classification (>III), (%)	98	98	98	1.00
Logistic EuroSCORE, (%)	24.72 ± 8.23	21.21 ± 8.76	25.17 ± 12.62	0.14
Previous pacemaker implantation, (%)	19	28	23	0.59
Pre-existing LBBB, (%)	8	7	16	0.29
Pre-existing RBBB, (%)	10	14	9	0.76
Pre-existing 1st degree AV block, (%)	5	26	14	0.03
Minimum diameter of LVOT, (mm)	18.72 ± 3.18	17.91 ± 2.79	19.02 ± 2.71	0.19
Maximum diameter of LVOT, (mm)	26.98 ± 3.09	26.90 ± 4.09	27.24 ± 2.87	0.89
Minimum diameter of aortic annulus, (mm)	19.89 ± 3.31	19.34 ± 2.26	20.33 ± 2.03	0.15
Maximum diameter of aortic annulus, (mm)	25.76 ± 3.24	25.23 ± 3.64	26.21 ± 2.04	0.33
Mean diameter of aortic annulus, (mm)	23.11 ± 2.82	22.31 ± 2.75	23.23 ± 1.84	0.18
Minimum diameter of sinotubular junction, (mm)	27.91 ± 3.34	27.01 ± 3.61	26.24 ± 3.59	0.30
Maximum diameter of sinotubular junction, (mm)	29.14 ± 3.36	28.83 ± 3.34	28.38 ± 3.70	0.75
Maximum diameter of ascending aorta, (mm)	34.25 ± 3.36	33.14 ± 3.35	32.22 ± 3.30	0.06
Angulation of aorta, (degrees)	44.60 ± 8.00	48.43 ± 9.75	46.42 ± 6.84	0.15
Calcium Score (mm^3^)	1403 (992, 1931)	1229 (890, 1738)	1654 (1190, 2108)	0.073

Values are expressed in their mean ± standard deviation/median (interquartile range) or as percentages. *p* Values in bold indicate statistical significance at 0.05 level. NYHA—New York Heart Association. LBBB—left bundle brunch block. RBBB—right bundle brunch block. AV—atrioventricular. LVOT—left ventricular outflow tract.

**Table 2 jcm-12-01739-t002:** Peri-procedural characteristics.

Characteristic	CoreValve^TM^	Evolut^TM^R	Evolut^TM^PRO	*p* Value
Balloon pre-dilatation, (%)	79	21	16	<0.01
Balloon post-dilatation, (%)	23	42	30	0.17
Pre-Implantation ID from NCC, (mm)	6.06 ± 1.11	5.22 ± 1.02	5.05 ± 1.02	<0.01
Implantation ID from NCC, (mm)	3.63 ± 2.77	4.96 ± 1.90	4.44 ± 2.24	0.07
Jump in NCC, (mm)	2.88 ± 2.33	1.48 ± 1.09	1.71 ± 1.35	0.01
ID over the annulus, (%)	23	5	9	0.02
Device success, (%)	98	100	100	0.99
PVL (none and grade 1), (%)	67	58	60	0.64

Values are expressed in their mean ± standard deviation or as percentages. *p* Values in bold indicate statistical significance at 0.05 level. ID—implantation depth. NCC—non-coronary cusp. PVL—paravalvular leak.

**Table 3 jcm-12-01739-t003:** Electrocardiographic changes.

Characteristic	CoreValve^TM^	Evolut^TM^R	Evolut^TM^PRO	*p* Value
LBBB immediately after TAVI, (%)	39	23	21	0.13
LBBB at 1st day after TAVI, (%)	30	21	12	0.13
RBBB immediately after TAVI, (%)	8	2	14	0.13
RBBB at 1st day after TAVI, (%)	3	2	2	0.98
1st AV block immediately after TAVI, (%)	33	12	12	0.02
1st AV block at 1st day after TAVI, (%)	22	30	9	0.05
Mobitz I after TAVI, (%)	0	0	0	1.00
Mobitz II after TAVI, (%)	0	2	0	0.38
Complete AV block immediately after TAVI, (%)	15	5	7	0.22
Complete AV block at 1st day after TAVI, (%)	0	5	0	0.16
Pacemaker implantation within the first day, (%)	33	19	7	0.01
Pacemaker implantation from day 1 to the point of discharge, (%)	2	2	5	0.77
Overall pacemaker implantation, (%)	38	19	9	<0.01

Values are expressed as percentages. *p* Values in bold indicate statistical significance at 0.05 level. TAVI—transcatheter aortic valve implantation. LBBB—left bundle brunch block. RBBB—right bundle brunch block. AV—atrioventricular.

## Data Availability

The data presented in this study are available on request from the corresponding author. The data are not publicly available due to privacy reasons.

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
