# Peer review of "Impact of Evolution of Self-Expandable Aortic Valve Design: Peri-Operative and Short-Term Outcomes"

_jcm, 2023, doi:10.3390/jcm12051739_

Round 1

Reviewer 1 Report

1) The severity of paravalvular leakage(PVL) was affected by the calcification burden of the aortic valve as the authors discussed. PVL was seemed to be lower in newer prosthetic valve in our clinical setting because newer valve had pericardial wrap. The authors should take into consideration the calcification burden of the aortic valve in enrolled patients. It will be better to show aortic valve calcium score to discus the result about the occurrence of PVL.

2) Implantation techniques were improved for these years to avoid the AV conduction disturbance. Why not the authors mention about this?

3) Did the authors analyze the length of membrane septum before procedure? Is it possible to add results about this?

Author Response

REVIEWER 1:

  • The severity of paravalvular leakage(PVL) was affected by the calcification burden of the aortic valve as the authors discussed. PVL was seemed to be lower in newer prosthetic valve in our clinical setting because newer valve had pericardial wrap. The authors should take into consideration the calcification burden of the aortic valve in enrolled patients. It will be better to show aortic valve calcium score to discus the result about the occurrence of PVL.

Reply: We really appreciate the Reviewer’s comment which aims at the improvement of the presentation of the quality of our work, and we apologize for this important omission regarding the calcification of the aortic valve.

In the revised manuscript and in the revised table 1 we provide data on total calcium score measurements (please see Table 1, page 4-5).

According to the revised data there was no difference in calcium score thought studied groups and PVL incidence and grade are unlikely to have been affected in the three studied groups by calcium score. This is discussed in the discussion section.

(please see page 7, lines 240-241).

We also revised accordingly the statistical analysis section

(please see page 3, lines 126-127 and 131-132).

In the methods section we also described how calcium score was calculated

(please see page 2, lines 86-90)

  • Implantation techniques were improved for these years to avoid the AV conduction disturbance. Why not the authors mention about this?

Reply: Again, we would like to thank the Reviewer for this important comment which aims at the improvement of our manuscript. This is now discussed in the revised discussion section

(please see page 7, lines 243-248).

  • Did the authors analyze the length of membrane septum before procedure? Is it possible to add results about this?

Reply: We really appreciate the importance of Reviewer’s comment about the need for description of the length of membranous septum, but unfortunately, we lack data on such measurements. However, we discussed this significant topic in the revised section of our manuscript.

(please see page 7, lines 244-248).

Reviewer 2 Report

In the present article, Dr. Evangelia Bei, et al demonstrated the impact of Core valve/Evolut series as one of the available TAVI devices to treat aortic valve stenosis.

Historically, self-expandable valve has a relatively high potential risk for pacemaker implantation after TAVI. This risk was a significantly lower trend because of the device development and physician’s learning curves.

The authors have reviewed and analyzed these Core valve/Evolut series and this would be meaningful for readers.

However, there are issues to consider:

Issues

·        To avoid pacemaker implantation after TAVI, the cusp-overlap technique would be beneficial technique. Probably this technique would affect these results. The authors should mention it.

·        The authors should mention future coronary intervention. The target implant depth would be higher in newer devices to avoid pacemaker implantation after TAVI. On the other hand, this consideration would result in difficulty for future coronary intervention. The authors should mention it in the discussion part.

Author Response

REVIEWER 2:

1) To avoid pacemaker implantation after TAVI, the cusp-overlap technique would be beneficial technique. Probably this technique would affect these results. The authors should mention it.

Reply: We really appreciate Reviewer’s helpful comment. According to the Reviewer’s suggestion, please see in the revised version of the manuscript the addition of (please see page 7, lines 268-262).

2) The authors should mention future coronary intervention. The target implant depth would be higher in newer devices to avoid pacemaker implantation after TAVI. On the other hand, this consideration would result in difficulty for future coronary intervention. The authors should mention it in the discussion part.

Reply: We would like to thank the reviewer for this valuable comment. Unfortunately, we have no data for additional coronary intervention after TAVI in the recorded patients.

Round 2

Reviewer 2 Report

This is well-revised.

I have no further comments.

Author Response

We thank the reviewer for the most welcome comments.
